# English- vs. Chinese-Medium Instruction in Chinese Higher Education: A Quasi-Experimental Comparison

**Haitao Guo [1], Fuhui Tong [2],\* 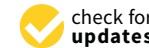, Zhuoying Wang [2], Yue Min [2] and Shifang Tang [2]**

[1]  College of Mass Communication, Shenzhen University, Shenzhen 518000, China; htguo@szu.edu.cn
[2]  Department of Education Psychology, Texas A&M University, College Station, TX 77840, USA;
    ustop2013wzy@email.tamu.edu (Z.W.); maya401@tamu.edu (Y.M.); shifangtang03@tamu.edu (S.T.)
\*  Correspondence: fuhuitong@tamu.edu

**Abstract:** Through a quasi-experimental approach, we compared Chinese college students' learning motivation, content knowledge, English language proficiency, and instructor's pedagogical practices between an English-medium instruction (EMI) and the parallel Chinese-medium instruction (CMI) course in a non-traditional discipline. Results indicated that EMI was more effective, as compared to CMI, in motivating students' learning of the focal subject. More specifically, EMI students held a stronger external goal orientation than did their CMI peers. Further, EMI and CMI students performed on par in their final exam in the subject and English after one semester of participation, controlling for their prior performance. The finding suggested that EMI did not carry a detrimental effect on Chinese college students' content area learning. Finally, observation revealed a significantly higher percentage of English language instruction focused on higher-order dense cognitive area in the EMI classroom where students were more engaged in their learning. Implications for policy and research were discussed regarding this educational approach for sustained and optimized student learning.

**Keywords:** English-medium-instruction; Chinese-medium-instruction; quasi-experiment; learning motivation; English proficiency; academic outcome; affective outcome

## 1. Introduction

At the turn of the century, China became a member of the World Trade Organization (WTO), a milestone of China's participation in the global economy. This has led to subsequent educational reform of internationalization in higher education as reflected in foreign language teaching and integrating English into content area instruction [1]. In the same year, the country's Ministry of Education [MOE] [2] issued guidelines to enhance quality of higher education by promoting the use of English-medium instruction (EMI) for at least 50% of the instructional time in content areas. A series of policies followed up to reinforce and sustain such reform (see [3,4]), resulting in a boom of EMI programs launched by Chinese higher education institutions. According to Yang and Zhang [5], there were 150–200 different EMI courses in some highly-ranked universities; an elite university in Beijing alone offered 200 EMI courses including those with 100% English instruction [6].

In contrast to the rising number of EMI courses in Chinese universities and a large body of work disseminated in Chinese language, scarce research has been conducted and published in English outlets to comprehensively examine these programs (e.g., [7,8]). For example, a systematic review of EMI research found only three studies from China that were written for an English-speaking audience [9]. Similarly, a recent meta-analysis in European context identified only five studies in English language that reported comparison between programs that used two languages as the medium of instruction and those that were monolingual on students' academic achievement [10]. Given such a small volume of work, the authors urged for more empirical publications in English language. Further, the literature

on EMI in China has been limited to policy documents (e.g., [11]), interviews (e.g., [12,13]), and survey data (e.g., [8,14]), underscoring a dire need of more data-driven research related to this topic [15,16]. Similarly, evidence of the advantage of EMI from a worldwide perspetive is still inconclusive after over a decade of research, mainly due to the lack of solid research methodology [9,17]. In response to this call, the present quasi-experimental study (see explanation in Section 3.1) joins the discussion through multiple lenses of observed instructor's practices, as well as college students' English language, academic, and affective outcomes. Our main objective is to examine differences between an EMI and its parallel Chinese-medium instruction (CMI) classrooms relating to these outcomes so as to determine the benefit of EMI and whether the quality of higher education can be enhanced through the educational and linguistic provision for sustained and optimized student learning.

In the remainder of this paper, we first present a review of the literature in Section 2 regarding student learning and pedagogical practices in EMI classrooms, followed by a description of research methods in Section 3. We then present quantitative and qualitative data analyses comparing EMI and CMI classes in the results section. These findings are discussed and commented on in Section 5. Our paper concludes with a section on limitations and implications of this study.

## 2. Literature Review

Subsumed under various forms of bilingual education, EMI is defined as an instructional practice in which English is used as the language of instruction in all subjects (such as finance, biology, and math [9,18]) except those that are teaching a specific language. With a growing global interest in bilingual education at the college level [19,20], EMI in China has been highly welcomed and accepted by higher education institutions [14]. It is argued that when students are exposed to content instruction delivered in their second language (L2, English in this study), its input becomes more meaningful for students to learn the target language incidentally [21,22] and to sustain their proficiency of academic English language [23]. It is worth noting that EMI does not exclude native language instruction, which is typical in many contexts, such as in China and Europe, and it is a highly recommended practice that, instead of replacing other languages, English should function in conjunction with these languages in EMI programs to address issues concerning language ideologies [24–26]. In this section, we review research related to student learning outcomes, classroom observation, and comparative studies in EMI programs.

### 2.1. Language and Academic Outcomes in EMI Programs

We located a handful of empirical studies published in Chinese reporting that EMI courses offered in computer science, geographic information, and welding technology were beneficial to students' English skills and subject knowledge [27,28]. For instance, Yan and Xu [27] confirmed that students' test scores increased not only in the content, but also in their English proficiency as a result of participating in EMI courses. In addition, students also held a favorable perception (i.e., positive attitudes) toward EMI courses offered in the disciplines of pharmaceutical biotechnology, history, and preventive medicine [29–31]. Take Han et al.'s study [29] as an example: over 80% of the EMI participants agreed that EMI was an effective approach for their achievement in both content knowledge and English listening and oral skills. Finally, Li [32] conducted correlation analyses on students' course achievement, English language proficiency, students' perceived satisfaction towards an EMI course, and students' workload caused by an EMI course. Participation in an EMI course was found to be positively associated with students' English language proficiency but not with academic performance.

A recent meta-analysis [22] revealed that EMI in Hong Kong secondary schools was most effective in promoting students' motivation engagement and English language proficiency, however, such advantage was not identified in either students' native language, or subject areas such as science, history, and geography. The research context of this work differs from our current study because in Hong Kong, English served as the language of society since the colonial period and continued

after the political handover in 1997, whereas in mainland China, English language has enjoyed its prestigious status as an international language since 1978 [33]; nevertheless, it calls for more empirical investigations into the effectiveness of EMI in mainland China on linguistic, academic, and affective outcomes, particularly in higher education where more EMI programs exist.

### 2.2. Students' Academic Motivation in EMI Programs

As an important area of research in educational psychology, motivation refers to people's psychological desires, beliefs, attitudes or intentions, which function as positive or negative simulations for guiding individual behavior or action [34]. According to self-determination theory (SDT), intrinsic motivation revolves around an inherent interest and enjoyment in participating in a task, whereas extrinsic motivation refers to engagement in learning that is driven by external factors [35]. Researchers have argued that extrinsic motivation has positive and significant value, especially when intrinsic needs are low [36,37]. It is acceptable to recognize and promote extrinsic motivation, particularly in Asian learning environments where societal and familial influences and feedback exert a substantive force to motivate college students' learning [38]. As a matter of fact, research showed that Chinese students' learning is characterized by extrinsic motivation among other features [39]. Liu [40] found that non-English majors were highly motivated to learn the language because higher English proficiency is related to a promising future and more career opportunities in China, which symbolized extrinsic learning motivation.

However, little scholarly attention has been drawn upon EMI participants' motivation, except the work conducted by Lasagabaster and colleagues among pre-collegial students in Spain [41]. For example, Lasagabaster and Doiz [42] examined secondary school students' affective factors as a result of participating in EMI programs. Results showed that both EMI and non-EMI groups demonstrated a similar level of motivation in learning English, while EMI learners were more intrinsically motivated than non-EMI learners in learning the subjects.

In the Chinese higher education context, students' attitudes toward EMI courses is the most frequently studied topic (e.g., [8,11,13,29,31]). Much less is researched, however, on students' experiences and involvement, such as self-identity [43] and learning motivation [44,45]. The only attempt to tackle Chinese college students' English learning motivation in EMI courses was reflected in two studies. Wei et al. [45] developed a questionnaire to investigate students' English learning motivational intensity in a less-privileged, lower-ranking university. Students generally agreed that EMI courses improved their English language proficiency. However, students' learning motivation or English learning effort was not found to be strong, with an overall of mean of 2.88 on a 5-point Likert scale. An earlier study using structural equation modeling identified a positive relationship between student English learning motivation and attitude toward EMI and such a relationship was moderated by students' major and English proficiency [44].

According to Lei and Hu [15] and Xu [14], the promotion and wide spread of EMI programs in mainland China have not been grounded in theoretical or empirical findings regarding the impact of EMI on students' learning motivation and beliefs. Comparative research reported that Chinese students' learning was mostly driven by societal and familial influences with pragmatic goals including higher grades, studying abroad, advancement in degree programs, and job placement and promotion, all of which reflected the Chinese cultural norm [46,47]. With the strong push of EMI programs by the central government, and the escalated status of English language that is associated with higher social rank and access to developed regions [33], we therefore hypothesize that students' learning motivation, particularly their extrinsic motivation, will be enhanced when enrolled in an EMI program which is associated with external influences.

### 2.3. Observation inside EMI Classrooms

In intervention research, systematic classroom observation has gained more popularity in the Western context because it generates more objective and reliable data of teachers' pedagogical

practices [48] and, therefore, has become a critical measure of fidelity of implementation that speaks to the quality of programs [49,50]. Nevertheless, observational research showed a mis-alignment between program label and actual classroom practice in terms of language distribution in English and students' native language (e.g., [51]). For this reason, it is particularly important to describe instructional practice when evaluating EMI programs.

To the best of our knowledge, there are only three empirical studies that evaluated the effectiveness of EMI programs in Chinese universities through observation in classrooms. For instance, in Tong and Tang's [52] study in a calculus EMI course, a majority of instructional time was allocated to teach content in Chinese, which failed to meet MOE's [4] requirement. Another study was performed in a comparison among EMI, CMI, and Chinese-English (CE) mixed instruction on teachers' questioning and student response [7]. It was reported that although there was no difference in, and low percentage of, higher-order questions across the classrooms, teachers asked lower-order thinking questions more frequently when they were teaching in EMI. The authors further identified that students' responses demonstrated a higher cognitive complexity when the language of instruction was Chinese instead of English. Finally, through a mixed approach of questionnaire, interview, and classroom observation, Yang [16] investigated the balance between content and language in EMI classrooms and found a predominant 'teacher talk' and little teacher-student interaction. Further, students did not favor mono-English instruction as it was not perceived to help them improve their English language proficiency or content knowledge.

## 2.4. Comparison between EMI and Non-EMI Courses

The literature reviewed above deals with an examination within EMI programs, however, without a fair comparison between EMI and non-EMI courses, no definitive and convincing conclusion can be drawn with a moderate internal validity. There are only three comparative studies published in English that we are able to locate, i.e., [11,15,53]. First, Han [53] carried out a quasi-experimental study of a college mathematics EMI course, which was found to be as equally effective as CMI in supporting students' academic achievement in mathematics, and was even more effective for students whose prior English was relatively low. Han emphasized that increasing the intensity of English in an EMI course had a positive effect on students' mathematics achievement as well as their motivation.

Second, Hu and Lei [11] compared EMI and CMI teachers' and students' beliefs, attitudes, and practices via one-on-one and focus group interviews. The findings indicated that there has been a gap between the goal of EMI policy and language practices in the classrooms and a misalignment between administrative support and actual needs of instructors and students. Finally, Lei and Hu [15] investigated the effect of instruction languages (English vs. Chinese) on students' English proficiency, learning motivation/attitudes towards English, and perception of EMI and language anxiety in an EMI and a parallel CMI program. Instructional language was not found to significantly impact students' English proficiency or English learning and use. However, EMI students' prior English proficiency and the perception toward EMI mediated the relationship between language of instruction and English language use and learning anxiety.

## 2.5. Critiques of Existing Literature and Rationale of the Study

In the aforementioned studies, researchers focused on one or two dimensions of the effect of EMI but have not systematically and comprehensively investigated such effects. A limitation in the three comparative studies is a lack of information of baseline equivalence either on English language proficiency (i.e., [11,15]), or mathematics achievement [53], which undermined the findings of quasi-experimental design, especially when these variables were the outcomes of study. Moreover, because in Han's [53] study no motivation instrument was involved to measure students' learning motivation of either English or mathematics, the statement of EMI motivating students was more of a speculation than a conclusion.

Taken together, the recent volume of research led by Chinese scholars uncovers more issues than solutions to improve the quality of EMI. This body of work reveals the following topics: (a) lack of competent EMI instructors as well as attention to the development of English language ([14,16]); (b) low level of English learning motivation [45]; (c) predominant Chinese language instruction particularly in higher-order questions, which significantly reduces the English instructional time that can be utilized to stimulate cognitive and critical thinking and engage students in academic English language [7,8]; (d) unachieved academic learning outcomes [15,32]; and (e) unbalanced distribution of resources to privileged and less-privileged universities [54]. Two non-exclusive searches among available literature related to EMI published in Chinese outlets revealed that a majority focused on theoretical elaborations or descriptive narration other than data-driven evaluation of EMI on students' English language and academic achievement [44,55]. A similar conclusion was summarized from a systematic review regarding EMI literature globally, namely, that there is a scarcity of rigorous research design to answer the question as to whether EMI is effective in promoting the outcomes [9]. It is also worth noting that the subject areas reviewed in the literature are mainly traditional and include a typical instructor-centered delivery and a final exam, such as in bio-chemistry (i.e., [8]), calculus [52], information technology [45], and business [11]. Little is known on EMI courses that are less-traditional and implement a different testing system to measure student learning.

These findings raise concerns regarding issues surrounding the quality of EMI and its further development and scale-up. One of the Sustainable Development Goals [56] is the quality of education to promote lifelong learning. Aligned with this agenda, in this study we seek to identify effective approaches to enhance motivation and learning of college students who are expected to contribute to the promotion of international communications and sustainability in global education in the field of EMI.

In this study, we conducted a quasi-experiment to compare college students' motivation, content knowledge, English language proficiency between EMI and parallel CMI courses in a non-traditional discipline. We further compared classroom practices between these two groups via a low-inference observation instrument. We also interviewed the EMI instructor to gain teachers' perception towards student learning in EMI programs. The following research questions were proposed:

1. Is there a difference between EMI and CMI college students on their learning motivation (including extrinsic motivation), course grades, and English exam, controlling for pre-test performance?
2. What is the time allocation of instruction in two languages (i.e., English and Chinese), content of instruction (i.e., dense cognitive), and students' communication modes (e.g., listening, speaking, reading, and writing) observed in EMI and CMI classrooms?
3. What is the instructor's perception of students' learning motivation and classroom participation, English proficiency, and performance on the subject in this EMI course?

## 3. Method

### 3.1. Research Context and Participants

This research was conducted in a public university located in a developed southern coastal area in China. The university was founded in late 1970s, at the time of China's economic reform and opening up to the global market. As the only multi-disciplinary university in the city, it is now serving more than 30,000 students with 2500 faculty and staff members. There are a number of government-endorsed EMI courses across disciplines, among which there is one national level EMI course in plant physiology and seven university-level EMI courses. This university is actively engaged in the economic expansion of the local area and nationwide, and has supported and fostered a number of provincial and national award programs, including in Mass Communication and Media. Therefore, there was an initiative to offer EMI courses in that program, which warranted a scientific investigation.



In this study, we used a quasi-experimental design which was defined as an experiment to estimate the impact of an intervention (in this case, EMI) on participants without a random assignment [57]. It is considered an alternative to true experimental design (which generates the most robust evidence to make causal argument), particularly when in reality it is not feasible to randomly assign participants to a particular condition. Indeed, participants in our study voluntarily registered for an EMI course or its parallel CMI course on the same subject taught by the same instructor. This instructor received a bachelor's degree from a top Chinese university, and a master's degree from a tier one institution in the United States. While pursuing a master's degree, he worked as a teaching assistant delivering English lectures and assisting undergraduate students with their projects for 2 years. To be qualified as a teaching assistant in that institution, an international student was required to pass an institutionalized English proficiency exam that measured reading, speaking, listening, and writing skills. Further, all international students were required to demonstrate sufficient English proficiency through standardized tests such as TOEFL and IELTS at the time of application. Therefore, this instructor was fluent in both academic English and mandarin Chinese.

In the fall of 2016 after completing a prerequisite CMI course of Television Art I, sophomores majoring in Mass Media were provided an opportunity to enroll in either EMI or CMI of Television Art II to be offered in the spring of 2017. Instead of random assignment, participation in the EMI/CMI class was completely voluntary with no minimum requirement/prerequisite. In total, there were 20 (8 males, 12 females) students enrolled in the EMI course and 27 (13 males and 14 females) in the CMI course that served as a comparison. Therefore, a quasi-experimental design was appropriate for this study after we evaluated the comparability/similarity of the two groups based on their demographic background listed in Table 1. Results from a *t*-test of students' age and years of English learning and from chi-square analysis of students' gender distribution and location of hometown indicated that there was no statistically significant difference between EMI and CMI students ($p > 0.05$). None of these students were exposed to any previous EMI programs.

**Table 1.** Demographic comparison between English-medium instruction (EMI) and Chinese-medium instruction (CMI).

| | Age | | Years of English Learning | | Gender Distribution | Location of Hometown |
|---|---|---|---|---|---|---|
| | **M** | **SD** | **M** | **SD** | **% Female (% Male)** | **(% from Developed Area)** |
| CMI | 21.6 | 0.87 | 12.4 | 2.3 | 52% (48%) | 78% |
| EMI | 21.6 | 0.63 | 13.2 | 1.2 | 56% (44%) | 93% |
| Sig level | 0.872 | | 0.203 | | 0.807 | 0.224 |

*3.2. Description of the EMI Course*

The 2-credit Television Art course was a core course intended to enhance students' theoretical understanding and practical production in film-making skills. The learning outcomes of this course included a thorough comprehension of the visualization process of scene and the visual language capabilities of long shots and montage, as well as the ability to selectively use long shots and montages to make and edit films. The instructional tools were Final Cut, PowerPoint, and DVD.

In the EMI course, all textbook and reading materials were in English. Students attended weekly classes for 3 h, for a total of 17 weeks and 51 h of study. At the beginning of the semester, the language distribution was 50% English and 50% Chinese; when the instructor observed that students were comfortable, he increased the percentage of English language instruction gradually to 100% toward the end of the semester. Chinese language was used occasionally to clarify terminologies. The grading system was based on 6 components with different weightings, i.e., attendance and classroom participation 5%, 4 field-based projects/presentations 15–20%, and final project/presentation on montage 25%. Students were formed into groups and required to use English in their presentations.

During each presentation, students in the same group received the same score, which was calculated based on a rubric developed by the instructor. In a typical EMI class, the instructor began the course by explaining new terminologies from the textbook, and introducing the topic of that class. The instructor then played a short classical movie clip with original English dubbing, followed by asking questions about the artistic strategies in film-shooting. English materials were then presented by the instructor, who explained and clarified specialized vocabulary and knowledge. Students were assigned into groups of 3–4 to discuss the newly learned theory and demonstrate understanding of the content by organizing their own language in English.

### 3.3. Description of the CMI Course

The 2-credit Television Art CMI course was designed equivalently to its EMI counterpart. The only difference was the language of instruction being 100% Chinese, and the textbook was a Chinese translation of the original English version. Due to the lack of translated reading materials, the instructor used the same English reading materials for instruction. The same grading system was used in the CMI class and students were also formed into groups for field-based assignments and presentations, which were conducted in Chinese. The structure of CMI resembles that of EMI described above.

### 3.4. Measure

- *Student academic learning motivation.* Motivated Strategies for Learning Questionnaire (MSLQ)-Chinese Adult Version [58,59] was used to measure students' learning motivation toward the course. It was translated, adapted, and validated from the original MSLQ developed by Pintrich and his colleagues [60] on undergraduate students' self-regulated learning in academic motivation and learning strategies of a specific discipline. Exploratory and confirmatory factor analyses were conducted in the validation of MSLQ-Chinese version for adult learners (MSLQ-CAL). This instrument used a 7-point Likert scale from "1 = not at all true of me" to "7 = very true of me" and was completed in a paper-pencil format. The Motivation scale in the MSLQ-CAL contained 5 factors: *Extrinsic Goal Orientation*, *Self-Efficacy for Learning and Performing*, *Task Value*, *Test Anxiety*, and *Control of Learning Beliefs*. Overall internal consistency (Cronbach's alpha) of the sample in this study was 0.87. Data were collected prior to the beginning of the semester and were treated as pre-test measure, and at the end of the semester as a post-test measure.

- *English proficiency.* To measure participating students' English language proficiency, we collected their scores of College English final examinations at the end of the previous fall semester (i.e., covariate) and the following spring semester (i.e., post-test) when the study took place. The final examination was developed by the English department in that university and administered to all non-English majors at the end of each semester. It followed the national English language curriculum (thus, with content validity) and was a 100-point test that included multiple choice questions, essay writing, cloze (grammar questions), and reading comprehension.

- *Academic achievement.* We used students' final grades from the spring semester as post-test, and their grades from the prerequisite course in the previous fall semester as a covariate.

- *Classroom observation.* To describe pedagogical characteristics and compare the instructional delivery between EMI and CMI classes, we used a low-inference, objective observation protocol, i.e., transitional bilingual observation protocol (TBOP) based on bilingual pedagogical theory [61]. TBOP was originally designed and validated for bilingual and ESL classrooms in the United States (e.g., [50,62]), and was also used by Tong and Tang [52] in their study within a college EMI calculus course in China. This four-dimensional instrument covers language of instruction (i.e., L1 which is Chinese in this study; L2 which is English), language content (e.g., lower-order thinking/interpersonal communication, and higher-order thinking/academic language), communication mode (e.g., reading, writing, speaking, and listening, or a combination of the four language skills), and teacher-student interaction (e.g., teacher asking questions/student

answering questions). Following the recommendation and practice of TBOP [50,62], both EMI and CMI classes were observed in 3 separate class periods across the semester through a high-definition audio recorder placed near the instructor to record his instruction as well as interaction with students. The recorded lessons were then coded by the researchers who were well trained in TBOP. Inter-rater reliability was established at Gwet $AC_1 \geq 0.6$, in alignment with the recommended process [50].

- *Teacher perception.* A virtual interview was conducted with the instructor regarding his perception of the EMI course delivery. We asked the instructor the following questions: (a) which class of students was more actively involved in oral activities and presenting their project; (b) what is your evaluation of EMI students' English proficiency; and (c) do you observe any difference between the two groups on their learning and mastery of the subject?

*3.5. Data Analysis*

Before comparing EMI and CMI courses regarding students' motivation, English language proficiency, and content learning, we examined baseline equivalence between students in the 2 courses with regard to these 3 outcomes using independent *t*-tests. Further, to address the first research question, we performed an analysis of covariate (ANCOVA) to examine the impact of EMI on students' motivation, English language proficiency, and their content learning, as compared to their CMI peers. To address our second research question on teachers' instructional practice, we conducted a chi-square test of homogeneity of proportion between EMI and CMI courses on the specific domains of TBOP. To answer our last research question, we reported results from the interview with the instructor on his perception of students' learning motivation, English proficiency, and performance in the course. SPSS 22.0 was used in the analyses of the first two questions, and effect size that quantifies statistical difference was reported in the form of Cohen's *d* and Cramer's V [63].

## 4. Results

Before addressing research questions 1–3, we examined baseline equivalence of students' academic achievement, English language proficiency, and learning motivation prior to their participation in either EMI or CMI. Descriptive statistics are displayed in Table 2. Results in Table 3 indicated that there was no statistical significance between EMI and CMI students regarding their pre-test performance in learning motivation, academic achievement, and English language proficiency. We then answered the research questions in the order they were proposed. Although all students participated throughout their course, data from two participants in EMI and two in CMI were invalid and, therefore, were removed from final analyses, resulting in 18 EMI students and 25 CMI students.

**Table 2.** Descriptive statistics of learning motivation, academic achievement, and English language proficiency by course.

| Measure | EMI (n = 18) | | | | CMI (n = 25) | | | |
|---|---|---|---|---|---|---|---|---|
| | Pre-Test | | Post-Test | | Pre-Test | | Post-Test | |
| | Mean | S.D. | Mean | S.D. | Mean | S.D. | Mean | S.D. |
| Learning motivation | 4.65 | 0.73 | 4.82 | 0.65 | 4.54 | 0.49 | 4.49 | 0.45 |
| Academic achievement | 76.83 | 9.14 | 78.17 | 4.41 | 75.67 | 8.01 | 75.93 | 6.52 |
| English language | 3.47 | 0.69 | 3.43 | 0.56 | 3.38 | 0.73 | 3.28 | 0.87 |

**Table 3.** Baseline equivalence on achievement of learning motivation, academic achievement, and English language proficiency.

| Measure | *t* | *df* | Sig. (2-Tailed) | Cohen's *d* |
|---|---|---|---|---|
| Learning motivation | 0.008 | 41 | 0.994 | 0.002 |
| Academic achievement | 0.370 | 28 | 0.715 | 0.14 |
| English language | 0.372 | 38 | 0.712 | 0.12 |

*Research Question 1. Is there a difference between EMI and CMI college students on their learning motivation, academic achievement, and English language proficiency, controlling for pre-test performance?*

Results from ANCOVA revealed that after adjustment for pre-test, there was a statistically significant difference in post-test learning motivation in favor of EMI, $F(1, 36) = 4.478$, $p = 0.041$, Cohen's $d = 0.71$. We further examined the effect of EMI on each factor of learning motivation and reported the descriptive statistics by course type in Table 4. Post hoc analyses suggested that that there was a statistically significant difference between EMI and CMI in the factor of *Extrinsic Goal Orientation*, $F(1, 36) = 8.216$, $p = 0.007$, Cohen's $d = 0.96$. Given multiple comparisons, Benjamini-Hochberg correction [64] was applied to control for type I error, yielding a new critical $p$ value of 0.01 ($0.05 \times 1/5$) for the factor of *Extrinsic Goal Orientation*. This number is still greater than the calculated $p$ value, therefore, the result remained statistically significant.

**Table 4.** Descriptive and inferential statistics of students' learning motivation in EMI and CMI by factors.

| Factors | EMI (n = 14) | | | | CMI (n = 25) | | | | Sig. | Cohen's $d$ | Cronbach's Alpha |
|---|---|---|---|---|---|---|---|---|---|---|---|
| | PRE-TEST | | Post-Test | | PRE-TEST | | Post-Test | | | | |
| | Mean | S.D. | Mean | S.D. | Mean | S.D. | Mean | S.D. | | | |
| Self-Efficacy for Learning & Performance | 4.7 | 1.0 | 4.8 | 0.8 | 4.6 | 0.8 | 4.6 | 0.6 | 0.392 | 0.29 | 0.85 |
| Extrinsic Goal Orientation | 5.2 | 1.4 | 5.7 | 0.8 | 4.6 | 1.0 | 4.8 | 0.8 | 0.007 | 0.96 | 0.64 |
| Task Value | 5.8 | 0.8 | 5.7 | 1.3 | 5.6 | 0.8 | 5.4 | 0.9 | 0.529 | 0.21 | 0.82 |
| Test Anxiety | 3.7 | 1.4 | 4.2 | 1.2 | 3.8 | 1.1 | 3.8 | 0.7 | 0.121 | 0.53 | 0.72 |
| Control of Learning Beliefs | 2.6 | 1.0 | 3.0 | 0.7 | 3.3 | 1.2 | 3.4 | 0.6 | 0.104 | 0.56 | 0.54 |

After adjustment for prior academic achievement, there was no statistically significant difference in post-test performance between EMI and CMI, $F(1, 37) = 3.905$, $p = 0.116$ (See Table 4). However, the mean score of EMI was numerically higher than that of CMI in the post-test of academic achievement (see Table 2). Similar results were identified in students' English language proficiency. After adjustment for prior English attainment, there was no difference in post-test English exam, $F(1, 37) = 4.406$, $p = 0.613$, although EMI students scored numerically higher than their CMI peers.

*Research Question 2. What is the time allocation of instruction in two languages (i.e., English and Chinese), content of instruction (i.e., dense cognitive), and students' communication modes (e.g., listening, speaking, reading, and writing) in EMI and CMI classrooms?*

In order to evaluate instructional activities in EMI and CMI classrooms, three major domains—language instruction, language content, and students' communication modes—were examined. The chi-square test detected a statistically significant association between type of instruction and teachers' instructional language, $\chi^2(4) = 112.4$, $p < 0.001$ with a large effect size (Cramer's V = 0.559). More specifically, the instructor spent more time in Chinese instruction in CMI (42.8%) than EMI (9.4%), (Cohen's $d = 1.66$), whereas there was a significant difference in the percentage of English instructional time (EMI = 54.4%, CMI = 11.7%, Cohen's $d = 1.7$). It is worth noting that the 9.4% Chinese instructional time in EMI was allocated to clarify the English terminologies. As we described earlier, observational data were collected at the beginning, middle, and end of the semester and the language distribution in the beginning was 50% English and 50% Chinese. Consequently, in the analyses in which data were aggregated, 9.4% reflected Chinese instruction that was observed during an earlier period of the semester in EMI. Further, English instruction only occurred when the English reading materials were being introduced in that language. Similar results were observed in students' language, $\chi^2(3) = 96.1$, $p < 0.001$ with a strong effect size (Cramer's V = 0.517). Post-hoc analyses revealed a statistically higher proportion of time when EMI students were speaking in English when provided the opportunity (CMI = 0%, EMI = 27.2%, Cohen's $d = 6.86$).

We further compared EMI and CMI classrooms through a cross-domain analysis of language content by language of instruction. We were particularly interested in how the instructor allocated instructional time in teaching dense cognitive higher-order thinking in EMI and CMI classes. Results

indicated that there were statistically significant differences between EMI and CMI in language content that was delivered in different languages ($p$ = 0.011, Cramer's V = 0.607). More specifically, 43.6% of the instructional time was allocated in teaching dense cognitive content (e.g., new knowledge with specialized vocabulary and procedure that stimulates students' higher-order/critical thinking) in CMI classroom, as compared to a 56.4% (Cohen's $d$ = 1.63) in English dense cognitive content in EMI classroom. Finally, in the domain of communication mode, it was observed that CMI students spent more time in reading (CMI = 16.1%, EMI = 0), while EMI students spent more time in listening-speaking (CMI = 1.7%, EMI = 12.2%, Cohen's d = 2.33). Very little teacher-student interaction was observed in CMI classroom.

*Research Question 3. What is the instructor's perception towards this EMI course?*

A virtual interview was conducted to investigate teacher's perception and reflections toward EMI and CMI courses. When asked which class of students was more actively involved in oral activities and in presenting their project, the instructor commented,

> *"EMI students were more serious and better prepared for the presentations and demonstrated a higher quality. It is probably because they feel the challenge of presenting and learning in English, and therefore spend more time on the coursework. They were also enthusiastic in active class participation."*

The instructor observed that in EMI class, students were more motivated by such challenges and demonstrated strong commitment and communication skills in both classroom activities and final presentation. Moreover, regarding his personal evaluation of students' English skills, the instruction shared,

> *"I was very surprised and impressed by students' English language proficiency. It was much higher than I expected. The stereotype of Chinese college students' relatively low English oral proficiency was not observed in this class. When provided more opportunities to practice in an EMI context, students' oral English proficiency can be improved."*

Finally, we inquired about students' subject area learning between the two classes. The instructor responded, *"Based on my observation and evaluation, both classes performed well on their field-based assignments. Overall, they have mastered the concept and skills very well."*

## 5. Discussion

Despite the popularity of EMI in Chinese higher institutions, limited empirical research has been conducted to explore the experiences of stake holders (i.e., students and teachers) [55]. With lack of sufficient information, there are mixed findings about the impact of this nation-wide educational policy on students' achievement of English language and disciplinary learning, as well as students' attitude and motivation towards EMI [15,52]. In this exploratory, quasi-experimental study, we attempted to generate some empirical evidence of the topic. Detailed discussions are presented below.

### 5.1. EMI vs. CMI: Motivational Outcome

Our hypothesis that EMI programs can enhance students' learning motivation in learning the subject was supported by the finding with a medium to large effect size. More specifically, EMI students held a stronger external goal orientation. They were more engaged in learning tasks and believed that the result of learning was contingent upon their commitment and effort put forth in this course. This finding was also reflected in the interview with the instructor, who commented that EMI students were more actively participating in the class and demonstrated higher communication skills in English.

Our finding is aligned with the socio-cultural perspective of motivational theory which posits that learners develop motivation through social support [65]. Given the elite status of English language that is associated with social up-mobility and access to the world [33,43], and driven by external forces

of parental and social expectation for job placement in foreign-run corporations and admission to graduate schools overseas [47], it is not surprising that students with more exposure and experiences with courses taught in English, or EMI, are more likely to develop a strong extrinsic goal orientation toward learning. These pragmatic goals have probably become a compelling force of learning that fits into the Chinese educational context and socio-cultural norm [38,47]. Further, our finding also echoes conclusions from previous studies in Taiwan, Hong Kong, and Spain (i.e., [22,42,66,67]) that learning content through a foreign/second language can promote motivation in learning because students were provided opportunities and support to practice their knowledge and understanding of the subject in English, and thus feel it is important to achieve a goal in the classrooms. More empirical support can be found in Korean higher education, a similar socio-educational context, where students expressed a strong interest in EMI courses, which were believed to help them in their study and overseas experiences, implying an extrinsic learning motivation [68].

One may argue that students with stronger motivation (e.g., it is popular to learn content through English) are more likely to enroll in the EMI course. However, the analysis of students' motivation at the beginning of the semester suggested that the two classes were equally motivated in learning this subject, but after one semester of participation, EMI students were even more externally motivated. Future research is recommended to capture such motivation alongside the motivation in learning English in EMI classrooms (e.g., [45]) and over a longer period of time (e.g., [42]), and the association between motivational and learning outcomes. We also recommend future research to explore approaches in EMI that promote students' intrinsic learning motivation, which has been consistently found to positively correlate with academic achievement and life-long learning [69,70]. Such information will contribute to the knowledge base and shed light on the design and implementation of EMI courses.

## 5.2. EMI vs. CMI: Academic and English Learning Outcome

Our results also showed that students in EMI class did not differ from their CMI peers on the final grades of the Television Art course and English exam after one semester, controlling for their performance prior to the start of the program. Such finding is consistent with the limited existing comparative studies that Chinese college students performed equally well on their subject area and English language learning, regardless of language of instruction (i.e., [15,53]). These findings can possibly be explained by two reasons. First, the in-house English achievement test was not sensitive to EMI because the test measured students' general English proficiency, whereas the EMI course was expected to improve students' specialized, content-specific English proficiency. Second, because the grading system was based on innovation that is unique to this non-traditional subject, students from the same group received the same score, which did not differentiate their subject area learning and therefore reduced the variation on their final grades. To the least extent, however, we failed to detect a detrimental effect of EMI in which the subject was taught in English for Chinese college students. Research in North America and Europe has supported a multi-year learning process of academic language in English for non-native speakers [62,71–74]. Because in our study students demonstrated a stronger extrinsic motivation, which is reported to benefit student learning [75,76], we are planning a longitudinal study to examine whether the extrinsic motivation can impact academic and language outcomes in EMI programs.

## 5.3. EMI vs. CMI: Observed Instructional Practices

Using a low-inference reliable instrument, we observed that more than half of the instructional time was delivered in English in the EMI classroom, and less than half of the instructional time was allocated to Chinese in the CMI classroom. EMI students' language usage also reflected an alignment with such a distribution and students were provided more opportunities in practicing oral English through presentation and discussion, and engaging in more student-teacher interactions. These findings are inconsistent with what was reported in current observational research that EMI teachers spent less than 10% in English delivery [8] and students were passive learners without opportunity to

pratice their oral English [16]. A lower percentage of Chinese instruction identified in CMI classroom is consistent with the finding that students spent more time in reading instructional materials or presenting their projects. Because of the nature of this course, teacher talk did not dominate lesson delivery in either EMI or CMI classrooms.

In addition to language of instruction, we further examined content of instruction in English, which revealed a higher percentage of instruction delivered in dense cogntive higher-order thinking area in English in EMI class, more so than the higher-order thinking area taught in Chinese in CMI class. This suggests that EMI instruction is conducive to students' learning of the subject, which is evidenced to support the development of students' academic language in English [50]. The finding stands in contrast to Hu and Li's [7] observation that teachers failed to engage students in cognitive challenging content in EMI classrooms. Therefore, the actual occurrence in this EMI course was in accordance with the expectation set by MOE [3] with English being the dominant langlue of instruction.

Through a systematic classroom observation evaluation on teachers' instructional languages and students' communication modes between EMI and CMI, this study extended Tong and Tang's [52] EMI classroom observation case study by providing a more comprehensive evaluation of EMI courses through a solid research design of quasi-experimental approach that has the potential of generating highly credible evidence. As Tong and Tang [52] pointed out, descriptions of teaching practices that clarifying how teachers scaffold EMI are still limited in China. We agree with Hu and Li [7] and recommend that EMI instructors need to be involved in the evaluation of their own instruction through peer-assisted or self-recorded classroom observation. More research of this theme will further the understanding of pedagogical practices in EMI classrooms, which lays a foundation for program evaluation.

## 6. Limitation

There are a few methodological limitations to external and internal validity of this study. First, it was conducted in a university located in an economically developed area of China. More research is needed to expand to less-privileged regions/universities for a broader generalizabilty. Second, there is a potential confounding factor of only one class in each condition. On a related note, there is also a low number of participants involved in this study. As a result, caution needs to be taken regarding the generalizability of our contextual findings. We recommend a larger sample of teachers (and students) with varying level of experiences and English proficiency that will generate more valuable and valid information, and future research is much needed to evaluate the effectives of EMI through a scientifically sound approach [9]. Third, this quasi-experimental study was implemented for only one semester. A longer term of implementation may generate more rigorous evidence of the effect of EMI on students' learning and motivation. Finally, literature shows that Chinese science majors differ from their social science peers in learning motivation and strategies [59]. The subject area of interest in this research is a non-traditional discipline and we posit that a comparison between different discplines can shed light on whether EMI can be effective for all students regardless of their field of study, when implemented with quality. Relatedly, given the nature of this focal discipline, academic outcome was assessed through instructor-developed assignments and its psychometric qualities in these teacher/institution-developed instruments are yet to be established. The same challenge applied to the English measure in this study. Therefore, results on these two outcomes should be interpreted with caution. Because it was the first time for this non-traditional EMI course to be offered, modifications to course design and delivery, and follow-up studies are planned to include more EMI and CMI classes, an individual final exam on the subject, and data collection on national standardized English tests to more closely investigate the impact of EMI programs on students' academic and English performance.

### 7. Implication for Policy and Research

Despite the above limitations, there are two implications that can be derived from the findings of this exploratory quasi-experimental study. First, although 17 years have passed since MOE announced the EMI/bilingual course policy and the majority of Chinese universities have launched numerous EMI courses in their undergraduate programs, the focal question as to whether such top-down, well-intended educational policy leads to favorable outcomes remains unanswered [15,32]. This is partly due to a lack of a commonly adopted, comprehensive evaluation framework that draws from, and is informed by, empirical evidence produced through quality research. Our study serves as a demonstration of such an evaluation and we purport that students' learning motivation, mastery of the subject knowledge, English language proficiency, and classroom pedagogical practices, as well as stakeholders' (e.g., students and instructors) perception, should be the key components of evaluation, among others. Equally important are well-designed experimental or quasi-experimental studies that are grounded by this framework to address the benefits of EMI programs with solid evidence. There should be a policy initiative in place to reinforce an evaluation system to be integrated into the design of EMI programs so as to ensure that students receive quality instruction that is motivating and promotes their learning.

Given the paucity of literature addressing EMI in higher education in China, the second implication calls for an open communication to enrich the discussion of EMI with non-Chinese academics and audiences. During our literature search, we identified a large number of studies disseminated within the Chinese academic discourse and, therefore, anticipate that an even more substantial body of work would emerge through a systematic search. In order to achieve active participation among the global scholarly community, which is aligned with the original intent of EMI, a thorough exploration into these studies is expected to promote a dialogue between the East and the West, and to contribute to the understanding of the effectiveness of EMI in a global context.

To conclude, one critical goal of education is to enhance sustainable development as reflected in students' life-long learning [56,77]. We believe that the educational initiative of EMI reinforced by the Chinese government is in line with this global agenda, and, therefore, worthy of further scientific investigation so as to improve the quality of higher education, which may inform practice and development nationally and internationally.

**Author Contributions:** Conceptualization, H.G. and F.T.; Data curation, H.G. and Z.W.; Formal analysis, Z.W., Y.M. and S.T.; Funding acquisition, H.G. and F.T.; Investigation, H.G.; Methodology, F.T.; Project administration, H.G. and F.T.; Writing—original draft, F.T., Z.W. and Y.M.; Writing—review & editing, H.G. and S.T.

**Funding:** The study was funded by Shenzhen University, Shenzhen, China; The open access publishing fees for this article have been covered by the Texas A&M University Open Access to Knowledge Fund (OAKFund), supported by the University Libraries and the Office of the Vice President for Research.

**Conflicts of Interest:** The authors declare no conflicts of interest.

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
