# Peer review of "English- vs. Chinese-Medium Instruction in Chinese Higher Education: A Quasi-Experimental Comparison"

_sustainability, doi:10.3390/su10114230_

Round 1
Reviewer 1 Report
It is an interesting and valuable study; its undoubted advantage is not much research in this field. The article can be improved and in particular:
The the main objective of the study must be precisely defined and exposed.
The introduction should be extended and a short information about the structure of the text should be added.
The authors should explain to the reader what the quasi-experiment is about and why they used this method in such studies. The "Method" section does not contain information on this subject.
The authors should stress the link between research and sustainability (as they aspire to publish in a specialist journal dealing with the issues of sustainable development).
Regarding the English language used in the paper, there is room for significant improvement. That is, the authors should try to include more linking words to combine their ideas, so that the reader can effortlessly progress from one idea to the next.
The results of analyzes should be more widely and clearly commented.
Author Response
Please see the attached letter. Thank you!

Reviewer 2 Report
LINE/SECTION | COMMENTS |
13 | Remove and from before English |
14 | Do you mean in a non-traditional? |
34 | Be sure to define tertiary education as not all countries use the same system |
44 | Remove to before a |
41 | Has there been an abundant amount of research done in Chinese outlets? This sentence makes it seem like there has been, but it isn’t discussed here |
45 | Of the benefit |
Entire paper | Double check paper requirements for correct method to cite references throughout manuscript |
47 | To what question are you referring? I can guess, but no clear question has been referenced yet |
47-48 | Rewrite sentence – awkward and unclear |
50 | Multitudes of lenses? |
54 | As the language |
54 | In non-language |
54 | What do you mean non-language subjects? So everything but those courses that are teaching a specific language? |
56 | At the college |
56 | Keep the tense consistent throughout – has been highly welcomed or is highly welcomed |
57 | Argued |
58 | Its |
63 | Do you really mean OR their subject knowledge? Not both? |
2.1 | This first paragraph is full of information, but most of it needs to be discussed specifically. For instance, provide more information about the favorable perceptions as that is unclear, discuss more about the first sentence, and define what you mean by perceptions. |
66 | Analyses |
68 | Study burden? |
Literature review | You have great information and sources about your field of study, but most of it is just stated and not really explained, making this section feel awkward and confusing. |
Entire manuscript | There are several spelling and grammar mistakes throughout the entire manuscript. Please have it doubled checked by a native speaker or English services to ensure it is correct. I will no longer include any spelling or grammar issues in this review. |
72 | What do you mean content areas? |
93 | Remember, this is international. Explain tier 2 university or reword |
95 | What do you mean by strong? To what? |
107 | If focusing on extrinsic motivation, should include a review of research done in the area of extrinsic motivation and be sure to explain what it is |
108 | Need to be more clear with why you hypothesize this and not just let readers speculate |
110-112 | Confusing sentence |
116-118 | What do you mean by this sentence? |
120 | Pedagogical occurrence? |
158-159 | Cite parts of this study to support your statements |
162-168 | Long sentence. Break apart or restructure to be more clear in your intent |
2.5 | First 2 paragraphs arer not about your study and thus do not belong in this section |
Research questions | These are not very specific, and do not focus on extrinsic motivation at all. They are vague and very broad. And e more clear with what you mean by instructor perception |
201 | Define focal university |
218 | What do you mean by minimal requirement? |
Table 1 | Be clear with how you calculated significance level |
233-236 | Your description of the EMI course doesn’t match your definition in the above reviews. This course was both English and Chinese, not just English. Either explain that this is normal for EMIs or you must change how you describe this course |
Method | Much of your descriptions are unnecessary. We don’t need to know the very specifics (like they watched The 3 little pigs), but those details that are important for us to understand the research and outcomes. |
329 | Are you reporting all the pretest performance results together here? Not needed, just refer to the table below |
Table 2 | Your n is different than the numbers you reported in the participants section. Is this drop due to students leaving the course or consent? Please explain. |
345 | Why the strange spacing? |
368 | This is not what you stated earlier when describing the course, when the instructor gradually moved to all English. Does the 9.4% include this time? |
389 | What were your interview questions? |
395-398 | Is the odd spacing between sentences due to cutting out parts of what was said? If so, you need to indicate that by using three dots “…” to show that you removed something that was said |
400 | Need quote to support your statement |
Discussion | Due to the very low number of student involved in this study, be VERY careful about generalizing your results. There should be no absolutes in your discussion |
416-417 | Unnecessary sentence |
423 | You need to include more literature review above about extrinsic motivation, especially that it is not considered the best type of motivation for learning. Your discussion makes it sound like this is a good thing, but the majority of research in this area shows it is not good for life long learning. |
5.2 | If numerically but not statistically higher, than it doesn’t matter what was observed or not – there is no difference. |
5.2 | You cannot write about the importance of EMI courses with data that shows them to show no statistical difference for language or academic outcomes. It is fine to discuss why, but do not try to defend the results or discuss as if there was a difference |
Manuscript | Did the instructor know of the study and what it was looking for? That can greatly impact what the instructor does in their course |
Limitations | Thank you for including them |
Author Response

(The authors gave the same response as above.)

Reviewer 3 Report
This is a useful paper on an important topic. I have a few comments to make:
In the introduction, you should make some reference to the large volume of research on EMI within Europe, particularly Spain. It is surprising that you do not mention authors such as Dafouz or Lasagabaster in this context. Other very influential authors within Europe are associated with groups in Maastricht (Wilkinson) or Jyvaskyla (Nikula), or Vienna (Dalton-Puffer).
In my view, bilingual education in the United States is NOT a relevant comparison, for several reasons: they are looking at bilingualism in early school years, they often regard bilingualism as negative, and the studies come from very different contexts from the university level EMI that you are studying. Europe would give you much closer comparisons.
The danger of self-selection is always present in CLIL/EMI research (see, for example, Rumlich on the "creaming effect". In order to address this possibility, you should give us more information about why the students chose EMI or CMI. Otherwise it is just a self-fulfilling prophecy - students who are more motivated to do English become more motivated when they do English.
In section 5.1 "affective" is not really the best word. I would say "motivational outcome", since this is what you talk about.
I am puzzled by the percentages of time in English/Chinese in section 5.3. Why is so little time spent in Chinese in the CMI classes? I think this needs more explanation.
Please revise the English. There are systematic problems with the definite article. Some sentences are unclear. You need to avoid contracted forms ("didn´t").
Author Response

(The authors gave the same response as above.)

Round 2
Reviewer 2 Report
Thank you for your changes and for being very clear with your responses to each concerns. Your revised manuscript looks great!